# How Hard Is Science?

**Adil Soubki** [1]   **Miles Cranmer** [1]

## Abstract

Symbolic Regression (SR) is the task of finding a closed-form mathematical expression that optimizes some objective. Solving this task is NP-hard. However, SR software routinely discovers accurate, interpretable models without exhaustively searching function space. Motivated by this disconnect between worst-case theory and practical success, we study SR through the lens of *parameterized complexity theory*. In particular, we reanalyze tractability with respect to practically relevant parameters like expression depth, tree size, and number of primitives used. We show that SR is fixed-parameter tractable (FPT) when parameterized by expression depth or tree size, formalizing the tractable regime exploited by bounded-complexity search in popular SR algorithms. However, SR becomes W[1]-hard when parameterized by the number of variables or primitives used, identifying selection as a source of intractability. We further find lower bounds under the exponential time hypothesis, prove approximation hardness, and rule out polynomial kernels for the case where the primitive set is part of the input.

## 1 Introduction

Newton begins the *Principia*'s "Rules of Reasoning in Philosophy" by declaring that "We are to admit no more causes of natural things, than such as are both true and sufficient to explain their appearances." In other words, keep it simple. In a few more words, one should keep the class of explanations *so simple* that finite evidence can rule most of them out. Without that restriction, fitting data is easy, since a more complicated explanation can always be cooked up, but interpretation becomes hopeless with so many possible answers to sift through. Newton was speaking of what he called "experimental philosophy" here,

but this can more-or-less be read as the first rule of doing science. This tradeoff between theoretical expressivity and empirical selection is inherited by symbolic regression.

Intuitively, symbolic regression (SR) is the problem of extracting predictive equations from data. Given some dataset $\mathcal{D} = \{(\mathbf{x}_i, y_i)\}_{i=1}^n$ where $\mathbf{x}_i$ is a vector of features, the idea is to discover an expression $f$ such that $f(\mathbf{x}) = y$, possibly accounting for noise. This workflow, where data is gathered and then used to build predictive models, has historically been done by hand. The canonical example is Kepler deriving his laws of planetary motion by painstakingly fitting equations to Tycho Brahe's astronomical records. In 1900, Planck did the same thing by matching a curve to blackbody radiation data before any quantum theory to explain it even existed.

Recently, the AI community has been interested in using machine learning to automate not just this process, but science more generally (Lu et al., 2024; Zheng et al., 2025; Gottweis et al., 2025). Implicit in this program is the assumption that mathematical science is computationally tractable. But how hard is science?

From a theoretical standpoint, SR appears intractable. Virgolin & Pissis (2022) proved that SR is NP-hard by reduction from unbounded subset sum, even for simple primitive sets. This suggests that finding optimal symbolic expressions should be computationally infeasible. Yet practical SR systems, despite all their limitations, *work*. They routinely discover accurate models in reasonable time (Schmidt & Lipson, 2009; Lemos et al., 2023). How can we resolve this apparent contradiction?

We approach this question by examining *what* makes SR hard. The NP-hardness proof of Virgolin & Pissis (2022) considers constructing expressions of possibly unbounded complexity to encode a difficult combinatorial problem.

| Problem Variant | Result | Reference |
|---|---|---|
| Depth ($k$, fixed $\mathcal{P}$) | FPT | Thm. 1 |
| Size ($s$, fixed $\mathcal{P}$) | FPT | Cor. 1 |
| Variables used ($k$) | W[1]-hard, $d^{k^{1-o(1)}}$ | Thms. 2 and 3 |
| Primitives used ($k$) | W[1]-hard, $|\mathcal{P}|^{k^{1-o(1)}}$ | Cor. 2, Thm. 3 |
| Approximation | Inapprox. | Thm. 4 |
| Kernelization ($\mathcal{P}$ as input) | No poly kernel | Thm. 5 |

*Table 1.* A summary of our results.

[1]Department of Applied Mathematics and Theoretical Physics and the Institute of Astronomy, University of Cambridge. Correspondence to: Adil Soubki <as3591@cam.ac.uk>.

*Proceedings of the 43rd International Conference on Machine Learning*, Seoul, South Korea. PMLR 306, 2026. Copyright 2026 by the author(s).

However, practical SR systems always impose complexity limits. For example, bounding the number of nodes in expressions or limiting the maximum number of function compositions a solution can contain. These heuristics used in practice change the computational problem.

To understand the complexity of SR under these constraints, we employ *parameterized complexity theory* (Downey & Fellows, 1999). Rather than measuring difficulty solely as a function of input size $n$, parameterized complexity analyzes runtime as a function of both $n$ and some parameter $k$ derived from the problem. This allows for a finer-grained classification. Two problems that are NP-hard in classical analysis may end up in different complexity classes under parameterizations. This framework is especially suited to SR where natural parameters abound. In this paper we map out this landscape to understand which parameters make SR tractable, and which leave it intractable. The results are summarized in Table 1.

The paper is organized as follows. We first provide some background on existing work in SR, complexity theory, and theoretical ML (§2). We then review definitions and concepts from parameterized complexity theory (§3). These tools are applied to examine parameterizations where SR is both tractable (§4) and intractable (§5). We then establish approximation hardness, showing that no FPT algorithm can achieve any finite approximation ratio for loss minimization even after avoiding zero-optimum instances (§6), and prove kernelization lower bounds demonstrating that efficient preprocessing is impossible for some parameterizations (§7). We conclude by discussing the implications of these results for algorithm design (§8) and some open questions (§9).

## 2 Background

**Symbolic Regression**  SR has been approached largely through two broad paradigms. The dominant approach uses genetic programming (GP), which evolves populations of expression trees through mutation and crossover (Koza, 1994; Poli et al., 2008; Orzechowski et al., 2018; La Cava et al., 2021). Modern GP-based systems like PySR (Cranmer, 2023) and Operon (Burlacu et al., 2020) achieve strong performance through careful engineering of genetic operators and complexity-bounded search. More recently, deep learning approaches have emerged. These include transformer-based methods like SymbolicGPT (Valipour et al., 2021) and NeSymReS (Biggio et al., 2021) that train on synthetic equation-data pairs, as well as reinforcement learning methods (Petersen et al., 2021) that learn to construct expressions sequentially. Hybrid approaches like AI Feynman (Udrescu & Tegmark, 2020) combine neural networks with physics-inspired decomposition strategies. Despite lacking formal guarantees, these systems routinely discover accurate models on scientific datasets.

See de Franca et al. (2025) for a list of the latest algorithms.

**Sparse Regression and Feature Selection**  The computational difficulty of selecting which operators and variables to include in SR is related to the notoriously difficult problem of feature selection (Kira & Rendell, 1992) in statistical modeling and its associated sparse regression variants. For example, it is known that computing a sparse solution to a linear system, even approximately, is NP-hard (Natarajan, 1995). This hardness has motivated work on convex relaxations like LASSO (Tibshirani, 1996) and modern mixed-integer optimization approaches (Bertsimas et al., 2016). Our W[1]-hardness result for $k$-variable SR (Theorem 2) shows that even an idealized SR setting inherits this selection hardness.

**Parameterized Complexity in ML**  The parameterized complexity paradigm has emerged as a useful tool for understanding ML algorithms. Existing results find that minimal decision tree learning is W[2]-hard when parameterized by tree size (Ordyniak & Szeider, 2021), though it becomes FPT under additional structural parameters (Gahlawat & Zehavi, 2024). Similar analyses have been applied to clustering (Cohen-Addad et al., 2019), neural network verification (Froese & Hertrich, 2023), and explainability problems (Ordyniak et al., 2024). Our work places SR in this landscape, with the finding that different parameterizations (structural vs. selection) yield different complexity classes.

## 3 Preliminaries

We now introduce the terms, definitions, and simplifications that will be assumed in subsequent sections.

### 3.1 Symbolic Regression

We follow the definitions of Virgolin & Pissis (2022) with minor modifications.

**Definition 1** (Primitive Set). A *primitive set* $\mathcal{P}$ consists of:

- *Functions $f : \mathbb{R}^{a_f} \to \mathbb{R}$ of arity $a_f \geq 1$*
- *Variables $x_1, \ldots, x_d$ (arity 0)*
- *Constants $c \in \mathbb{R}$ in finite precision (arity 0)*

We denote by $a = \max_{f \in \mathcal{P}} a_f$ the maximum arity. For our purposes, we will assume that $\mathcal{P}$ has finitely many elements.

*Remark* 1 (Numeric Encoding). Constants and data values are encoded as rationals in binary, with bit-length contributing to the input size. Our running-time statements use a unit-cost real-RAM model for primitive evaluation. Equivalently, in a bit-complexity model they require an a priori polynomial bound on intermediate numeric growth and on the cost of evaluating each primitive. For our hardness reductions, constructed constants have bit-length polynomial in the input size.

**Definition 2** (Expression and Search Space). An *expression*

over $\mathcal{P}$ is a rooted tree where internal nodes are labeled by functions and leaves by variables or constants, with each node's children matching the arity of its label. The *search space* $\mathcal{F}$ is the set of all such expressions.

**Definition 3** (Depth and Size). The *depth* of an expression is the length of the longest root-to-leaf path (a single node has depth 0). The *size* is the total number of nodes.

**Problem 1** (Symbolic Regression). Given a primitive set $\mathcal{P}$, dataset $\mathcal{D} = \{(\mathbf{x}_i, y_i)\}_{i=1}^n$, and loss function $\mathcal{L} : \mathbb{R}^n \times \mathbb{R}^n \to \mathbb{R}_{\geq 0}$, the SR problem asks for:

$$f^\star = \underset{f \in \mathcal{F}}{\arg\min} \, \mathcal{L}(\mathbf{y}, f(\mathbf{x}))$$

where $\mathcal{F}$ is the search space defined by $\mathcal{P}$.

We will focus on the decision version of the SR problem.

**Problem 2** (SR-DEC). Given $\mathcal{P}$, $\mathcal{D}$, $\mathcal{L}$, and threshold $\varepsilon \geq 0$, decide whether there exists $f \in \mathcal{F}$ with $\mathcal{L}(\mathbf{y}, f(\mathbf{x})) \leq \varepsilon$.

As in prior work (Virgolin & Pissis, 2022), we will assume that the function being discovered is not recursive. We also assume candidate evaluation and comparison of the loss with $\varepsilon$ are polynomial-time operations under the numeric model in Remark 1. We write $n$ for the number of data points, and use $\mathrm{poly}(n)$ as shorthand for polynomial dependence on the encoded input length, including the data values and the description of the loss.

### 3.2 Parameterized Complexity

We provide a brief introduction for readers unfamiliar with parameterized complexity; see Downey & Fellows (1999); Flum & Grohe (2006) for a comprehensive treatment.

Classical complexity theory measures the difficulty of a problem as a function of input size $n$. A problem in P admits algorithms running in time $n^{\mathcal{O}(1)}$, while NP-hard problems are believed to not admit polynomial-time algorithms. This binary classification is often too coarse and many NP-hard problems become tractable when some structural parameter of the input is small (Downey & Fellows, 1995b;a).

Parameterized complexity theory refines this picture by introducing a secondary quantity, the *parameter* $k$, and analyzing runtime as a function of both $n$ and $k$. We will focus on only a few of the complexity classes from this framework.

**Definition 4** (Fixed-Parameter Tractability). A parameterized problem is *fixed-parameter tractable* (FPT) if it can be solved in time $f(k) \cdot n^{\mathcal{O}(1)}$ for some computable function $f$, where $n$ is the input size and $k$ is the parameter.

The function $f(k)$ may grow rapidly with $k$, even as fast as a tower of exponentials, but the main point is that $n$ appears only polynomially. When $k$ is small, an FPT algorithm is efficient regardless of how large $n$ becomes. This captures the intuition that some problems are "easy when the parameter is small."

Not all parameterized problems are FPT. Just as NP-hardness provides evidence that a problem lacks polynomial-time algorithms, there is a hierarchy of complexity classes for parameterized problems. The most important for our purposes is the class W[1].

**Definition 5** (Class W[1]). A parameterized problem is in W[1] if it reduces to $k$-CLIQUE, the problem of deciding whether a graph contains a clique of size $k$, via an FPT reduction (a reduction running in time $f(k) \cdot n^{\mathcal{O}(1)}$ that maps parameter $k$ to parameter $k' = g(k)$). A problem is W[1]-hard if $k$-CLIQUE is FPT-reducible to it.

The problem $k$-CLIQUE is W[1]-complete (both in W[1] and W[1]-hard). Despite decades of effort, the best known algorithm runs in time $n^{\mathcal{O}(k)}$, and it is widely believed that no $f(k) \cdot n^{\mathcal{O}(1)}$ algorithm exists. The assumption $FPT \neq W[1]$ plays a role analogous to $P \neq NP$ in classical complexity. Under this assumption, W[1]-hard problems admit no FPT algorithms.

The classical analysis of SR by Virgolin & Pissis (2022) implicitly assumes that the size, depth, and primitive use of expressions may all grow with the dataset size $n$, but this is not very realistic. In practice, the number of primitives is limited by our mathematical vocabulary, and the expression size and depth are limited by a requirement for ease of human understanding. In this sense, it's more practical to treat these parameters as fixed rather than as functions of $n$. In the sections that follow, we reanalyze the problem with these parameters in mind.

## 4 When SR is Tractable

We establish that SR is FPT for natural complexity parameters. These results formalize the intuition that bounding expression complexity makes SR tractable in the regime where the primitive set is fixed.

### 4.1 FPT Parameterized by Depth

**Problem 3** (SR-DEC Parameterized by Depth). Given $\mathcal{P}$, $\mathcal{D}$, $\mathcal{L}$, threshold $\varepsilon \geq 0$, and parameter $k \in \mathbb{N}$: decide whether there exists $f \in \mathcal{F}$ with $\mathrm{depth}(f) \leq k$ and $\mathcal{L}(\mathbf{y}, f(\mathbf{x})) \leq \varepsilon$.

**Theorem 1** (SR is FPT by Depth). *Fix a primitive set $\mathcal{P}$ with maximum arity $a \geq 1$. Then SR-DEC parameterized by composition depth $k$ is FPT, solvable in time $(|\mathcal{P}| + 1)^{\mathcal{O}(N_k)} \cdot \mathrm{poly}(n)$, where $n = |\mathcal{D}|$ and $N_k$ is the size of the complete $a$-ary tree of depth $k$.*

*Proof.* We enumerate all expressions of depth at most $k$. The complete $a$-ary tree of depth $k$ has at most

$$N_k = \sum_{i=0}^{k} a^i = \begin{cases} \frac{a^{k+1}-1}{a-1} & \text{if } a > 1 \\ k+1 & \text{if } a = 1 \end{cases}$$

nodes. Every expression of depth $\leq k$ can be embedded into this complete tree by introducing a "null" label $\perp$ for unused positions. We enumerate all $(|\mathcal{P}|+1)^{N_k}$ labelings and filter to syntactically valid expressions (correct arities, no null labels in the actual expression). This is an overcount, but the total number of candidate labelings is at most $(|\mathcal{P}|+1)^{N_k}$.

For each valid expression $f$, we evaluate $f(\mathbf{x}_i)$ for all $i = 1, \ldots, n$. This requires $\mathcal{O}(N_k)$ primitive operations per data point under the numeric model of Remark 1. Computing the loss $\mathcal{L}$ takes $\text{poly}(n)$ time.

The algorithm enumerates all candidate labelings, filters to valid expressions, evaluates each, and returns YES if any achieves loss $\leq \varepsilon$. Total runtime:

$$(|\mathcal{P}|+1)^{N_k} \cdot \mathcal{O}(N_k \cdot n + \text{poly}(n)) = (|\mathcal{P}|+1)^{\mathcal{O}(N_k)} \cdot \text{poly}(n)$$

Since $\mathcal{P}$ is fixed, $|\mathcal{P}|$ and $a$ are constants. Thus $f(k) = (|\mathcal{P}|+1)^{N_k} \cdot N_k$ depends only on $k$, giving runtime $f(k) \cdot \text{poly}(n)$ as required for FPT. $\square$

The FPT result treats $|\mathcal{P}|$ as a constant, not part of the input. This models the regime where the operator vocabulary and feature dimension are fixed in advance, while the number of data points may grow. If the operator library is fixed but $d$ grows with the instance, enumeration gives a runtime which is FPT in $(k, d)$ but not in $k$ alone. The variable-selection results study this high-dimensional feature-selection regime.

### 4.2 FPT Parameterized by Tree Size

Many SR systems used in practice restrict the search by setting their measure of complexity to be tree size (number of nodes) and bound their search to some maximum complexity. While it is a slightly different parameterization, this also yields FPT.

**Problem 4** (SR-DEC Parameterized by Size). Given $\mathcal{P}$, $\mathcal{D}$, $\mathcal{L}$, threshold $\varepsilon \geq 0$, and parameter $s \in \mathbb{N}$: decide whether there exists $f \in \mathcal{F}$ with $\text{size}(f) \leq s$ and $\mathcal{L}(\mathbf{y}, f(\mathbf{x})) \leq \varepsilon$.

**Corollary 1** (SR is FPT by Size). *Fix a primitive set $\mathcal{P}$ with maximum arity $a$. Then SR-DEC parameterized by expression tree size $s$ is FPT, solvable in time $|\mathcal{P}|^{\mathcal{O}(s)} \cdot \text{poly}(n)$.*

*Proof.* It suffices to enumerate all expression trees of size at most $s$. For each $r \leq s$, the number of expression trees of size $r$ is bounded by the product of tree shapes and labelings. For trees with maximum arity $a$, the number of distinct tree shapes with $r$ nodes is at most $C_a^r$ for some constant $C_a$ depending only on $a$ (e.g., bounded by the $(r-1)$-th Catalan number for binary trees). Each shape admits at most $|\mathcal{P}|^s$ labelings. Thus the total count is at most $s \cdot C_a^s \cdot |\mathcal{P}|^s = s \cdot (C_a \cdot |\mathcal{P}|)^s$.

The enumeration and evaluation proceed as in Theorem 1. Since $\mathcal{P}$ is fixed, both $C_a$ and $|\mathcal{P}|$ are constants, giving

runtime $f(s) \cdot \text{poly}(n)$ for $f(s) = s \cdot (C_a \cdot |\mathcal{P}|)^s$. $\square$

## 5 When SR is Hard

The FPT results above are relevant when the measure of solution complexity being restricted is related to the size of the expression. In practice, this is often what is done. However, we can also consider parameterizations that restrict other reasonably natural notions of complexity. For example, suppose we want an expression which does not use too many primitives or incorporates a sparse set of variables. Since this seemingly reduces the search space, it is tempting to think that it would make the problem somehow easier, but this is not the case. In fact, when we must *select* which primitives or variables to use, SR becomes intractable.

### 5.1 W[1]-Hardness for Number of Variables

We first show that selecting which input variables to use makes SR hard, even when the only operator is addition. This connects SR to the well-studied feature selection problem (Kira & Rendell, 1992). In the optimistically restricted case of linear regression, sparse subset selection is NP-hard (Natarajan, 1995). This hardness persists in our parameterizations, even in the simplest SR settings.

**Problem 5** ($k$-Variable SR-DEC). Given $\mathcal{P}$, $\mathcal{D}$ over $d$ input variables, $\mathcal{L}$, threshold $\varepsilon \geq 0$, and parameter $k \in \mathbb{N}$: decide whether there exists $f \in \mathcal{F}$ using *exactly $k$ distinct input variables* with $\mathcal{L}(\mathbf{y}, f(\mathbf{x})) \leq \varepsilon$.

**Theorem 2** (SR-DEC is W[1]-hard by Variables Used). *$k$-Variable SR-DEC is W[1]-hard.*

*Proof.* We reduce from EXACT COVER, which is W[1]-complete when parameterized by the number of sets in the solution.[1]

**Exact Cover.** Given a universe $U$, a set $\mathcal{S} = \{S_1, \ldots, S_m\}$ of nonempty subsets of $U$, and parameter $k$: decide whether there exist exactly $k$ sets in $\mathcal{S}$ that partition $U$ (i.e., are pairwise disjoint and cover all elements).

**Reduction.** Given an EXACT COVER instance $(U, \mathcal{S}, k)$, we construct an SR instance as follows:

- **Variables:** For each set $S_j \in \mathcal{S}$, create a variable $x_j$. This gives $d = m$ variables.
- **Primitive set:** $\mathcal{P} = \{+\} \cup \{x_1, \ldots, x_m\}$. (Variables $x_1, \ldots, x_m$ are available as leaf nodes.)
- **Data points:** For each element $e \in U$, create a data point $(\mathbf{x}^{(e)}, y^{(e)})$ where:
  $x_j^{(e)} = 1$ if $e \in S_j$, and $x_j^{(e)} = 0$ otherwise.
  $y^{(e)} = 1$.

---

[1]The W[1]-completeness of EXACT COVER follows from W[1]-completeness of PERFECT CODE (Cesati, 2002) via a straightforward reduction; see also Downey & Fellows (1999).

This gives $n = |U|$ data points.
- **Loss:** Squared error, $\mathcal{L}(f) = \sum_{e \in U}(f(\mathbf{x}^{(e)}) - y^{(e)})^2$.
- **Threshold:** $\varepsilon = 0$ (exact fit required).
- **Parameter:** $k$ (number of variables to select, same as EXACT COVER parameter).

**Correctness.** We show that EXACT COVER has a solution iff the SR instance has a $k$-variable expression with zero loss.

($\Rightarrow$) Suppose $\{S_{j_1}, \ldots, S_{j_k}\} \subseteq \mathcal{S}$ is an exact cover of size $k$. Consider the expression $f(\mathbf{x}) = x_{j_1} + x_{j_2} + \cdots + x_{j_k}$, which uses exactly $k$ distinct variables. For any element $e \in U$, exactly one of the selected sets contains $e$ (since it's an exact cover), so exactly one term equals 1 and the rest equal 0. Thus $f(\mathbf{x}^{(e)}) = 1 = y^{(e)}$ for all $e$, achieving zero loss.

($\Leftarrow$) Suppose expression $f$ uses exactly $k$ distinct variables and achieves zero loss. Since the only operator is addition and all inputs are in $\{0, 1\}$ with target 1, we need $f(\mathbf{x}^{(e)}) = 1$ for all $e \in U$. The expression $f$ computes a weighted sum $f(\mathbf{x}) = \sum_{j=1}^{m} c_j x_j$ where $c_j \geq 0$ is the number of times variable $x_j$ appears. Let $T = \{j : c_j > 0\}$ be the variables used; by assumption $|T| = k$.

For any $e \in U$: $f(\mathbf{x}^{(e)}) = \sum_{j:e \in S_j} c_j = 1$. Since each selected set is nonempty, every $j \in T$ appears in the constraint for at least one element. Each $c_j$ is a positive integer, and for every element the sum over sets containing that element equals 1. Hence each selected set has coefficient $c_j = 1$, and each element is contained in exactly one selected set. Thus each element is covered by exactly one of the $k$ selected sets, forming an exact cover of size $k$.

The reduction runs in polynomial time and preserves the parameter, completing the proof. $\square$

*Remark* 2. This result identifies *feature selection* as the computational bottleneck in SR. Even when restricted to linear expressions, selecting which $k$ variables to include is W[1]-hard. Allowing more expressive forms (e.g., polynomials, compositions) cannot make the problem easier.

A concrete example of the reduction from EXACT COVER is given in Appendix A.1.

## 5.2  W[1]-Hardness for Number of Primitives

The hardness of variable selection extends to primitive selection. Since variables are primitives (they are arity-0 elements of $\mathcal{P}$), constraining the number of primitives is at least as restrictive as constraining the number of variables.

**Problem 6** ($k$-Primitive SR-DEC). Given $\mathcal{P}, \mathcal{D}, \mathcal{L}$, threshold $\varepsilon \geq 0$, and parameter $k \in \mathbb{N}$: decide whether there exists $f \in \mathcal{F}$ using *exactly $k$ distinct primitives* with $\mathcal{L}(\mathbf{y}, f(\mathbf{x})) \leq \varepsilon$.

**Corollary 2** (SR-DEC is W[1]-hard by Primitives Used).

*$k$-Primitive SR-DEC is W[1]-hard.*
*Proof.* We use the reduction from Theorem 2, which maps EXACT COVER instances to SR instances. In that reduction, $\mathcal{P} = \{+\} \cup \{x_1, \ldots, x_m\}$, and any zero-loss expression must be a sum of variables corresponding to an exact cover. To avoid an edge case, we may assume $k \geq 2$ since $k = 1$ instances of EXACT COVER are trivial.

A YES instance of EXACT COVER (exact cover of size exactly $k$ exists) yields an SR expression using exactly $k$ variables plus the addition operator, giving exactly $k + 1$ distinct primitives. Conversely, if an SR expression uses exactly $k + 1$ primitives and achieves zero loss, it must use the operator $+$ and exactly $k$ variables, which (by Theorem 2) corresponds to an exact cover of size $k$.

Thus $k$-EXACT COVER reduces to $(k + 1)$-Primitive SR-DEC. Since EXACT COVER is W[1]-complete, $(k + 1)$-Primitive SR-DEC is W[1]-hard, and hence so is $k$-Primitive SR-DEC by reparameterization. $\square$

Note that, in contrast to earlier, here we use "exactly $k$" rather than "at most $k$" in our problem definitions. This creates a small divergence in our analysis. Fortunately, the W[1] results can be lifted to the at-most-$k$ case without too much trouble.

**Corollary 3** (At-Most-$k$ Variable SR is W[1]-hard). *The variant of $k$-Variable SR-DEC that asks for an expression using at most $k$ distinct input variables is W[1]-hard.*

*Proof.* Reduce from the standard W[1]-hard restriction of EXACT COVER in which all sets are nonempty. Given an instance $(U, \mathcal{S}, k)$, add fresh elements $z_1, \ldots, z_k$. For each original set $S_j$ and each $r \in [k]$, create a variable $x_{j,r}$ corresponding to the set $S_j \cup \{z_r\}$. The data points are defined as before, with one point for each element of $U \cup \{z_1, \ldots, z_k\}$ and target value 1.

If the original instance has an exact cover $S_{j_1}, \ldots, S_{j_k}$, choose a bijection between these sets and the dummy elements and use the variables $x_{j_r,r}$. This gives a zero-loss expression using at most $k$ variables. Conversely, suppose a zero-loss expression uses at most $k$ variables. Since the only operator is addition, the expression computes,

$$f(\mathbf{x}) = \sum_{j,r} c_{j,r} x_{j,r}$$

with coefficients $c_{j,r} \in \mathbb{N}$ equal to variable multiplicities. On the dummy row for $z_r$, the target is 1, and exactly the variables $x_{j,r}$ evaluate to 1. Hence $\sum_j c_{j,r} = 1$ for every $r \in [k]$. Therefore, for each $r$, exactly one variable $x_{j,r}$ is selected, it has coefficient 1, and no selected variable is repeated. Thus the expression uses exactly $k$ variables, one for each dummy element. Because all original sets are nonempty, two selected variables with the same original index $j$ would over-cover at least one element of $U$. Hence

the selected variables correspond to $k$ distinct original sets, and restricting them to $U$ gives an exact cover. $\square$

While the proof is for the $k$-variable parameterization, the argument above easily translates to the $k$-primitive parameterization. Since the proofs are simpler when dealing with the exact-$k$ problem, we prefer that formulation but note that they are interchangeable. An example of this argument is in Appendix A.2 for reference.

Prior work frequently notes that SR systems operate best on low-dimensional, tabular data (Kamienny et al., 2022; Chen et al., 2017a). These results formalize why SR systems succeed with small, curated primitive sets but struggle with high-dimensional feature selection.

### 5.3 Lower Bounds via ETH

The W[1]-hardness results above rule out FPT algorithms (assuming $FPT \neq W[1]$), but do not specify *how* the runtime must depend on the parameters. The Exponential Time Hypothesis (ETH) provides more fine-grained lower bounds. ETH states that 3-SAT on $n$ variables cannot be solved in $2^{o(n)}$ time (Impagliazzo & Paturi, 1999). Our reductions allow us to make use of recent work that established ETH-based lower bounds for EXACT COVER (Guruswami et al., 2025).

**Theorem 3** (ETH Lower Bound for Selection). *Assume ETH. Then the following holds.* [2]

1. *$k$-Variable SR-DEC requires time $\Omega_k(d^{k^{1-o(1)}})$.*
2. *$k$-Primitive SR-DEC requires time $\Omega_k(|\mathcal{P}|^{k^{1-o(1)}})$.*

*Proof.* Guruswami et al. (2025) prove that, under ETH, $(k, \rho k)$-EXACT COVER (distinguishing instances with an exact cover of size $k$ from those where no $\rho k$ sets cover the universe) requires time $\Omega_k(m^{k^{1-o(1)}})$ for some constant $\rho \geq 1$, where $m$ is the number of sets. Since any algorithm for exact $k$-EXACT COVER also solves this gap version, the same lower bound applies to the exact problem.

The reduction in Theorem 2 produces $d = m$ variables and preserves $k$, so (1) follows. Part (2) follows from (1) via Corollary 2. $\square$

Since the brute-force algorithm runs in $\mathcal{O}(d^k)$ time, this means that the naive algorithm is (disappointingly) nearly optimal.

*Remark* 3 (Sparsity Relaxation). Inspired by this $\rho$-approximate version of EXACT COVER, instead of requiring exactly $k$ variables (or primitives) we could allow up to $\rho k$ for some constant $\rho \geq 1$. This models the scenario where we seek a sparse solution but would accept one that is slightly less sparse than optimal. This is distinct from

---

[2]The notation $\Omega_k$ indicates that the implicit constant in the $\Omega$ bound may depend on $k$.

loss approximation (Section 6). Here we relax sparsity, not fit quality. However, the lower bound above applies directly so even accepting solutions that are less sparse than optimal does not help.

## 6 Approximation Hardness

So far, we have studied variations on the *decision* problem SR-DEC where given a threshold $\varepsilon$, we must determine whether an expression achieving loss $\leq \varepsilon$ exists. In practice, we often care about the *optimization* problem. Instead of searching for a loss below some threshold, we ask for an expression that minimizes loss, or at least comes close to minimizing it.

**Problem 7** (SR Optimization). Given primitive set $\mathcal{P}$, dataset $\mathcal{D}$, and loss function $\mathcal{L}$, the *SR optimization problem* asks for an expression $f$ minimizing $\mathcal{L}(f, \mathcal{D})$. We denote the minimum achievable loss by

$$\mathrm{OPT} := \min_{f \in \mathcal{F}} \mathcal{L}(f, \mathcal{D}).$$

For parameterized variants, $\mathrm{OPT}_k$ denotes the minimum loss achievable by expressions satisfying the parameter constraint (e.g., using exactly $k$ primitives).

When exact optimization is intractable, we may settle for *approximation*: finding an expression with loss at most $\alpha \cdot \mathrm{OPT}$ for some approximation ratio $\alpha \geq 1$.

### 6.1 Inapproximability of Loss Minimization

**Definition 6** (FPT Approximation). An $\alpha$-FPT-approximation for exact-$k$ SR is an algorithm that, given any instance, outputs an expression $\hat{f}$ satisfying the exact-$k$ parameter constraint with $\mathcal{L}(\hat{f}, \mathcal{D}) \leq \alpha(k) \cdot \mathrm{OPT}_k$, running in time $f(k) \cdot \mathrm{poly}(n)$.

Our W[1]-hardness reductions have a special structure. By adding one unfittable point and amplifying the remaining points, we can make the optimal loss positive while preserving an arbitrarily large multiplicative gap.

**Theorem 4** (No Finite FPT Approximation). *For exact-$k$ Variable SR with sum-of-squared-error loss, there is no $\alpha(k)$-FPT-approximation for any computable approximation ratio $\alpha$, unless $FPT = W[1]$, even restricted to instances with $\mathrm{OPT}_k \geq 1$.*

*Proof.* Assume there is an $\alpha(k)$-FPT-approximation for the variable case. We reduce from EXACT COVER using the construction in Theorem 2, but modify the data. Let $R$ be an integer with $R > \alpha(k)$, which can be computed as a function of $k$. Duplicate every original data point $R$ times, and add one additional data point $(\mathbf{0}, 1)$, where all variables evaluate to $0$ and the target is $1$.

If the EXACT COVER instance is a YES instance, the corresponding sum of exactly $k$ variables fits all duplicated original points and misses only the new point, so $\mathrm{OPT}_k = 1$. If the source instance is a NO instance, every exact-$k$-variable expression makes an error of at least 1 on at least one original point. Because that point was duplicated $R$ times, and every expression also misses $(0, 1)$, we have $\mathrm{OPT}_k \geq R + 1$.

Run the assumed approximation algorithm and compute the loss of its output. On YES instances, the output has integer-valued squared loss at most $\alpha(k) \cdot 1 < R$, hence at most $R - 1$. On NO instances, every feasible expression has loss at least $R + 1$. Thus the approximation algorithm distinguishes YES from NO instances in FPT time, contradicting W[1]-hardness unless $FPT = W[1]$. □

The proof also holds for exact-$k$ Primitive SR under the corresponding reparameterization (set $q = k + 1$ and choose $R > \alpha(q)$ instead).

When $\mathrm{OPT}_k = 0$, any finite approximation ratio requires finding an exact solution. This is a degenerate gap, which is why the theorem above avoids the case. An example walking through this argument is given in Appendix A.3. It's worth mentioning that this lower bound should be thought of as a worst-case obstruction. It shows that approximation alone does not eliminate combinatorial difficulty but does not preclude useful approximation guarantees under additional assumptions.

## 6.2 A Note on Constants

Practical SR systems treat constants differently from our formal model. Rather than selecting constants from a finite set, they often consider constants to be placeholder variables that are optimized for each candidate structure. This raises the question of whether the complexity results change under this *optimizable constants* model.

The reductions in Theorems 2 and 4 and Corollary 2 use primitive sets containing only $+(\cdot, \cdot)$ and variables. Since no constants are used in the arguments, the W[1]-hardness and inapproximability results hold regardless of how constants are modeled. The FPT enumeration results survive exactly when the inner problem of fitting constants for each candidate structure can itself be solved in $g(k)\mathrm{poly}(n)$ time, or is supplied as an oracle with that cost. General nonlinear constant optimization need not satisfy this requirement (Cai & Chen, 1997; Chen et al., 2017b), and practical local optimizers do not by themselves provide global guarantees.

## 7 Kernelization Lower Bounds

Even for parameterized problems, we might hope for efficient preprocessing that compresses the whole instance (including the data) to size depending only on the parameter $k$. For the fixed-primitive formulation, bounded size gives an FPT enumeration algorithm. If the primitive set is instead part of the input, bounded size no longer yields the same fixed-parameter behavior, and in fact the instance cannot be polynomially compressed in the size parameter unless standard complexity collapses occur.

### 7.1 Background: Kernelization

**Definition 7** (Kernelization). A *kernelization* for a parameterized problem $Q$ is a polynomial-time algorithm mapping any instance $(x, k)$ to an instance $(x', k')$ such that $(x, k) \in Q \Leftrightarrow (x', k') \in Q$ and $|x'|, k' \leq g(k)$ for some computable function $g$. If $g$ is polynomial, we say $Q$ has a *polynomial kernel*.

A polynomial kernel would compress the whole SR instance to $\mathrm{poly}(k)$ size. We show that this cannot be done for the primitive-set-as-input formulation considered in this section. To do this, we will borrow an existing result.

**Definition 8** (OR-Cross-Composition (Bodlaender et al., 2014)). An *OR-cross-composition* of a language $L$ into a parameterized problem $Q$ is a polynomial-time algorithm that takes $t$ instances $x_1, \ldots, x_t$ of $L$ (equivalent under some polynomial equivalence relation) and outputs $(y, k')$ where:

(i) $(y, k') \in Q \Leftrightarrow \exists i : x_i \in L$
(ii) $k' \leq \mathrm{poly}(\max_i |x_i| + \log t)$

**Theorem** (Bodlaender et al. (2014)). *If an NP-hard language $L$ OR-cross-composes into parameterized problem $Q$, then $Q$ has no polynomial kernel unless* NP $\subseteq$ coNP/poly.

### 7.2 No Polynomial Kernel for SR

We OR-cross-compose from an NP-hard restriction of UN-BOUNDED $k$-SUM in which $k \leq |S|$. [3]

**Unbounded $k$-Sum.** Given a finite set $S = \{a_1, \ldots, a_n\}$ of positive integers, target $T$, and integer $k \leq |S|$: decide if there exist $k$ elements from $S$ *with repetition allowed* that sum to $T$. Equivalently, do there exist non-negative integers $m_1, \ldots, m_n$ such that $\sum_{i=1}^n m_i = k$ and $\sum_{i=1}^n m_i a_i = T$?

The connection between SR and unbounded subset sum was established by Virgolin & Pissis (2022) for NP-hardness. We extend their construction using cross-composition machinery to rule out polynomial kernels for the primitive-set-as-input formulation of SR.

**Theorem 5** (No Polynomial Kernel). *For the formulation in which $\mathcal{P}$ is part of the input,* SR-DEC *parameterized by expression size $s$ has no polynomial kernel unless* NP $\subseteq$ coNP/poly.

---

[3]This restricted version of UNBOUNDED $k$-SUM is NP-hard via reduction from EXACT COVER. Encode each set as a base $B = k + 1$ number representing membership with target requiring each element to be covered exactly once. See Claim 1.

*Proof.* Use the polynomial equivalence relation that groups instances by their common value of $k$. On the restricted source language, this is a polynomial equivalence relation because $k \leq |S|$. Take $t$ equivalent instances $(S_1, T_1, k), \ldots, (S_t, T_t, k)$, where $S_i = \{a_{i,1}, \ldots, a_{i,n_i}\}$. Let $A = \max_{i,j} a_{i,j}$.

**Element Tagging.** Choose an integer base $B > \max_i T_i + kA$. Define tagged values:

$$\tilde{a}_{i,j} = a_{i,j} + B^i.$$

The tag $B^i$ uniquely identifies which instance element $a_{i,j}$ came from.

**SR Instance Construction.** Construction is as follows.

- **Primitives:** $\mathcal{P} = \{+\} \cup \{c_{i,j} : i \in [t], j \in [n_i]\}$ where constant $c_{i,j}$ has value $\tilde{a}_{i,j}$.
- **Data:** $t$ data points with dummy input $\mathbf{x}$ and targets:
$$y_i = T_i + k \cdot B^i \quad \text{for } i = 1, \ldots, t$$
- **Loss:** 0-1 loss (number of mispredictions), threshold $\varepsilon = t - 1$.
- **Parameter:** Expression size $s = 2k - 1$.

The threshold $\varepsilon = t - 1$ means "at least one data point must be predicted correctly."

**Correctness ($\Rightarrow$).** Suppose instance $m$ has a solution: a selection of $k$ elements (with repetition) from $S_m$ summing to $T_m$.

The corresponding expression $F$ (sum of $k$ tagged constants from instance $m$) has size $2k - 1$ and evaluates to $T_m + k \cdot B^m = y_m$. This correctly predicts $y_m$, achieving loss $t - 1 \leq \varepsilon$.

**Correctness ($\Leftarrow$).** Suppose expression $F$ with size $\leq 2k-1$ achieves loss $\leq t - 1$.

Then $F$ correctly predicts some $y_m$. Since $F$ has size $\leq 2k - 1$, it is a sum of at most $k$ constants (with possible repetition). Let $s_i$ denote the count of constants selected from instance $i$ (counting multiplicity), so $\sum_i s_i \leq k$. Then:

$$F = \sum_{(i,j) \in \text{sel}} a_{i,j} + \sum_{i=1}^{t} s_i \cdot B^i$$

where the first sum counts elements with their multiplicities. For $F = y_m = T_m + k \cdot B^m$, the untagged sum satisfies

$$\sum_{(i,j) \in \text{sel}} a_{i,j} \leq kA < B,$$

and also $T_m < B$. Thus there is no carry between the units place and the $B^i$ tag digits. Base-$B$ representations are unique, so matching $T_m + kB^m$ requires $s_m = k$ and $s_i = 0$ for all $i \neq m$. Therefore all selected constants come

from instance $m$, exactly $k$ of them are selected, and their untagged values sum to $T_m$, giving a valid UNBOUNDED $k$-SUM solution for instance $m$.

**Parameter Bound.** Since the source language has $k \leq |S_i|$, we have $k \leq \max_i |x_i|$. Hence the combined parameter $s = 2k - 1$ is polynomial in $\max_i |x_i| + \log t$. The construction is polynomial in the total input size.

By Section 7.1, SR-DEC has no polynomial kernel unless $\mathsf{NP} \subseteq \mathsf{coNP/poly}$. $\square$

An example of the reduction used in this argument is in Appendix A.4 for reference.

Practically, this result says that, when the primitive library is part of the input, there is no polynomial-time preprocessing that compresses the entire SR instance to size polynomial in the expression-size parameter alone. This contrasts with problems like VERTEX COVER, which admits a kernel with $2k$ vertices (Nemhauser & Trotter Jr, 1975; Fellows et al., 2011). This lower bound is separate from the fixed-primitive-set FPT regime of Theorem 1 and Corollary 1.

*Remark* 4. If we return to the model used in Theorem 1 and Corollary 1, there are combinations of $\mathcal{L}$ and $\mathcal{P}$ that, in fact, *do* admit polynomial kernels. Consider, for example, a language which allows for only polynomials with squared loss. However, this is not true for all such languages. Adding an operator that squares ($\text{sq}(f) = f^2$) breaks this construction.

## 8 Discussion

There is a fairly clean dichotomy in the parameterized complexity of SR presented here. When the parameter bounds the size of the search space, either by tree depth or size, SR is fixed-parameter tractable (Thm. 1 and Cor. 1). When the parameter requires selection from a possibly large set of options, SR becomes W[1]-hard (Thm. 2 and Cor. 2).

This suggests that the difficulty of SR does not lie solely in the challenge of enumerating a large solution space but is, at least partially, inherited from the difficulty of the feature selection problem.

### 8.1 Practical Implications

These results support several empirical observations about the behavior of SR systems.

**Heuristics** Systems like PySR (Cranmer, 2023) and Operon (Burlacu et al., 2020) bound expression size (typically $\leq 30$ nodes) or depth ($\leq 6$) to restrict the space being explored. With fixed primitive sets of typically no more than 10–20 operators and the assumption that the chosen constant optimizers will yield fits that are "good enough", this closely matches our constructions in the FPT regime

(Theorem 1 and Corollary 1). GPs explore this space heuristically rather than exhaustively but the similarity remains. Some work has actually explored explicitly enumerating small expression spaces as a viable approach, aligning even more strongly with the results studied here (Bartlett et al., 2024), and additional analysis has compared this somewhat favorably with other search strategies (Kronberger et al., 2024).

**The Feature Selection Bottleneck** Existing empirical work on SR frequently cites the challenges current systems face when searching for simple expressions in high-dimensional data (Dong & Zhong, 2025; Bartlett et al., 2025). Despite these difficulties, Chen et al. (2017a) note that few modern systems integrate feature selection before learning into their pipeline. Our W[1]-hardness results (Theorem 2 and Corollary 2) provide theoretical context for these observations and suggest that integrating feature selection is a promising direction. Furthermore, it is likely that automatically discovering useful primitives is a fundamentally more challenging problem without relaxations.

## 8.2 Connection to Program Synthesis

SR can be thought of as a restricted form of *program synthesis* (Gulwani et al., 2017), where the "program" is a symbolic expression valid in the grammar generated by $\mathcal{P}$ and the specification is the dataset $\mathcal{D}$. Our FPT results align with the idea of introducing *syntactic bias* by restricting the domain-specific language in order to help scale methods. The work presented could likely be rephrased to provide bounds for program synthesis as well. To our knowledge, there are few parameterized analyses in this area.

## 8.3 Limitations

The definition of SR used throughout this paper, of course, does not capture all aspects of the problem.

**Noise** For one, it is implicitly assumed that minimizing loss on the data to achieve an exact fit ($\varepsilon = 0$) is the ideal solution. In reality, data contains noise that should be modeled, making the desirability of solutions more a question of user intent. We conjecture that our hardness results are robust to noise.

**Constants** As mentioned previously, our model assumes constants are drawn from a finite set. In some ways, this is realistic as computers typically represent numbers in fixed-precision. In others, it is an idealization since most algorithms employ some form of optimizer to fit constants.

**Worst-Case Complexity** Our hardness results are worst-case. To achieve them, we construct adversarial instances that may be unlikely to occur in physical laws or natural data distributions. Imposing additional structure to account for this would give a more complete picture of the behavior of practical SR systems.

## 9 Conclusion

We applied parameterized complexity to study SR in detail. Our results show that in a parameterized setting, hardness stems from the combinatorial selection required to find sparse solutions rather than from the size of the search space alone. We are also able to extend some of these results to get ETH lower bounds and to the setting of approximation, independently of the method used for handling constants. Finally, we demonstrate that no polynomial kernel exists for the primitive-set-as-input size parameterization by reduction from UNBOUNDED $k$-SUM. This provides theoretical context for the practical success seen by so many of the existing algorithms for SR. In fact, the parameters examined in our FPT proofs are often small, resulting in a tractable regime for such cases when the primitive set is fixed in advance.

**Open Problems** We list a few ideas for continuations of this work in the future.

- **Tighter FPT bounds:** Is the $|\mathcal{P}|^{O(a^k)}$ algorithm for depth-$k$ SR optimal?
- **Average-case complexity:** Is SR easier on "natural" data distributions arising from physical laws?
- **Positive approximation:** Are there restricted settings (e.g., monotone primitives, bounded coefficients) where FPT approximation is possible?
- **Modeling constants:** Can we get more exact results for the various approaches to handling constants?
- **Data kernels:** Which fixed primitive languages and losses allow for efficient data compression?

## Impact Statement

This paper presents work whose goal is to advance the field of Machine Learning. There are many potential societal consequences of our work, none which we feel must be specifically highlighted here.

## Acknowledgments

The authors would like to thank our anonymous reviewers for their perceptive comments, which improved this work. Adil Soubki thanks the UK Science and Technology Facilities Council (STFC) for a Ph.D. studentship. Miles Cranmer is grateful for support from the Schmidt Sciences AI2050 Early Career Fellowship and the Isaac Newton Trust.

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

# A   Reduction Examples

We walk through some concrete examples of the reductions done in Theorems 2, 4 and 5 and Corollaries 2 and 3.

## A.1   EXACT COVER to $k$-Variable SR

We illustrate the reduction from Theorem 2 with a concrete instance.

Consider the universe $U = \{1, 2, 3, 4, 5, 6\}$ with four candidate sets: $S_1 = \{1, 2, 3\}$, $S_2 = \{4, 5, 6\}$, $S_3 = \{1, 2, 4\}$, and $S_4 = \{3, 5, 6\}$. The EXACT COVER question asks whether two of these sets partition $U$. In this case the answer is yes, since $S_1 \cup S_2 = U$ and $S_1 \cap S_2 = \emptyset$.

The reduction constructs an SR instance as follows. We introduce one variable $x_j$ for each set $S_j$, giving four variables in total. The primitive set is simply $\mathcal{P} = \{+, x_1, x_2, x_3, x_4\}$. For each element $e \in U$, we create a data point whose features encode set membership: $x_j^{(e)} = 1$ if $e \in S_j$ and 0 otherwise. The target for every data point is $y^{(e)} = 1$. The resulting dataset is:

| Element $e$ | $x_1$ | $x_2$ | $x_3$ | $x_4$ | Target |
|:---:|:---:|:---:|:---:|:---:|:---:|
| 1 | 1 | 0 | 1 | 0 | 1 |
| 2 | 1 | 0 | 1 | 0 | 1 |
| 3 | 1 | 0 | 0 | 1 | 1 |
| 4 | 0 | 1 | 1 | 0 | 1 |
| 5 | 0 | 1 | 0 | 1 | 1 |
| 6 | 0 | 1 | 0 | 1 | 1 |

We use squared-error loss with threshold $\varepsilon = 0$, and ask whether a two-variable expression achieves zero loss.

The exact cover $\{S_1, S_2\}$ corresponds to the expression $f(\mathbf{x}) = x_1 + x_2$. Evaluating this on each data point yields output 1 everywhere: for element 1, we have $x_1 = 1$ and $x_2 = 0$, so $f = 1$; for element 4, we have $x_1 = 0$ and $x_2 = 1$, so again $f = 1$; and similarly for the other elements. The expression uses exactly two variables and achieves zero loss.

Now consider the non-cover $\{S_1, S_3\}$. These sets overlap (both contain elements 1 and 2) and fail to cover elements 5 and 6. The corresponding expression $f(\mathbf{x}) = x_1 + x_3$ produces:

| Element $e$ | $x_1 + x_3$ | Target | Error |
|:---:|:---:|:---:|:---:|
| 1 | 2 | 1 | 1 |
| 2 | 2 | 1 | 1 |
| 3 | 1 | 1 | 0 |
| 4 | 1 | 1 | 0 |
| 5 | 0 | 1 | 1 |
| 6 | 0 | 1 | 1 |

Elements 1 and 2 are double-covered (output 2 instead of 1), while elements 5 and 6 are missed entirely (output 0). The total squared error is 4, not zero.

The pattern is general. For any selection of variables $T \subseteq \{1, 2, 3, 4\}$, the expression $\sum_{j \in T} x_j$ evaluated at element $e$ counts how many of the selected sets contain $e$. Achieving output 1 for every element requires that each element belong to exactly one selected set. This is precisely the definition of an exact cover.

## A.2   EXACT COVER to At-Most-$k$ Variable SR

We illustrate the reduction from Corollary 3 with a concrete instance.

We reuse the universe $U = \{1, 2, 3, 4, 5, 6\}$ and the sets $S_1, S_2, S_3, S_4$ from Section A.1, again asking whether two of these sets partition $U$. As before the answer is yes, since $S_1 \cup S_2 = U$ and $S_1 \cap S_2 = \emptyset$. The point of this reduction is to make hardness survive when the algorithm is permitted to use *fewer* than $k$ variables: dummy elements force any zero-loss expression to use exactly $k$ variables anyway.

The reduction adds $k = 2$ fresh elements $z_1, z_2$ to the universe. For each pair $(j, r)$ with $j \in \{1, 2, 3, 4\}$ and $r \in \{1, 2\}$, we introduce a variable $x_{j,r}$ corresponding to the augmented set $S_j \cup \{z_r\}$, giving $mk = 8$ variables in total. The primitive set is $\mathcal{P} = \{+\} \cup \{x_{j,r} : j \in [4], r \in [2]\}$. The dataset has one point per element of $U \cup \{z_1, z_2\}$ with target $y^{(e)} = 1$:

| Element $e$ | $x_{1,1}$ | $x_{1,2}$ | $x_{2,1}$ | $x_{2,2}$ | $x_{3,1}$ | $x_{3,2}$ | $x_{4,1}$ | $x_{4,2}$ | Target |
|---|---|---|---|---|---|---|---|---|---|
| 1 | 1 | 1 | 0 | 0 | 1 | 1 | 0 | 0 | 1 |
| 2 | 1 | 1 | 0 | 0 | 1 | 1 | 0 | 0 | 1 |
| 3 | 1 | 1 | 0 | 0 | 0 | 0 | 1 | 1 | 1 |
| 4 | 0 | 0 | 1 | 1 | 1 | 1 | 0 | 0 | 1 |
| 5 | 0 | 0 | 1 | 1 | 0 | 0 | 1 | 1 | 1 |
| 6 | 0 | 0 | 1 | 1 | 0 | 0 | 1 | 1 | 1 |
| $z_1$ | 1 | 0 | 1 | 0 | 1 | 0 | 1 | 0 | 1 |
| $z_2$ | 0 | 1 | 0 | 1 | 0 | 1 | 0 | 1 | 1 |

The rows exhibit a clean duality. On an original element $e \in U$, the entry $x_{j,r}^{(e)}$ depends only on $j$ (since $z_r \notin U$), so the two columns for each $j$ agree on the top six rows. On a dummy $z_s$, the entry depends only on $r$, with $x_{j,r}^{(z_s)} = 1$ exactly when $r = s$, so the bottom two rows have 1s only in the columns matching their own $r$.

We use squared-error loss with threshold $\varepsilon = 0$, and ask whether an expression using at most two variables achieves zero loss.

The exact cover $\{S_1, S_2\}$ pairs with the dummies via the bijection $r = 1 \mapsto S_1$ and $r = 2 \mapsto S_2$, yielding the expression $f(\mathbf{x}) = x_{1,1} + x_{2,2}$. Evaluating on each row produces 1 everywhere: elements 1, 2, 3 contribute through $x_{1,1}$ alone; elements 4, 5, 6 through $x_{2,2}$ alone; dummy $z_1$ matches $x_{1,1}$ but not $x_{2,2}$; dummy $z_2$ matches $x_{2,2}$ but not $x_{1,1}$. The expression uses exactly two variables and achieves zero loss.

Now suppose we try to economize and use only a single variable. Take $f = x_{1,1}$. It evaluates correctly on elements 1, 2, 3 and on the dummy $z_1$, but produces 0 on every other row. Even though the at-most-$k$ formulation permits an expression with one variable in principle, the dummy $z_2$ alone forces nonzero loss. More generally, achieving zero loss on the row $z_s$ requires at least one selected variable with $r = s$, so the dummies force at least $k$ variables to be used.

The other failure mode is selecting two variables with the same original index $j$. Consider $f(\mathbf{x}) = x_{1,1} + x_{1,2}$, which covers both dummies but draws both selections from $S_1$:

| Element $e$ | $x_{1,1} + x_{1,2}$ | Target | Error |
|---|---|---|---|
| 1 | 2 | 1 | 1 |
| 2 | 2 | 1 | 1 |
| 3 | 2 | 1 | 1 |
| 4 | 0 | 1 | 1 |
| 5 | 0 | 1 | 1 |
| 6 | 0 | 1 | 1 |
| $z_1$ | 1 | 1 | 0 |
| $z_2$ | 1 | 1 | 0 |

Elements 1, 2, 3 are double-covered (the same original set is selected twice) while 4, 5, 6 are missed entirely. This is precisely the failure mode the "all sets nonempty" assumption rules out: the elements of $S_1$ witness the duplication, and the argument would break if $S_1$ could be empty.

The pattern is general. The dummies force one variable per $r \in [k]$, giving exactly $k$ variables in any zero-loss expression. The nonempty-sets assumption then forces these $k$ variables to have distinct original indices, and restricting them to $U$ recovers an exact cover of size $k$.

## A.3  Approximation Hardness via Loss Amplification

We illustrate the construction from Theorem 4, which adapts the $k$-variable reduction (Theorem 2) to rule out FPT approximation. The primitive set, variables, and parameter are inherited from that reduction; only the dataset is modified.

The construction begins with a computable approximation ratio $\alpha$ that we wish to rule out and chooses an integer $R > \alpha(k)$ as a function of $k$. For concreteness, take $\alpha(k) = k$ and $k = 2$, so $\alpha(k) = 2$ and we may pick $R = 3$. The modification is then: *(i)* duplicate every data point produced by the reduction of Theorem 2 three times, and *(ii)* append a single new data point $(\mathbf{0}, 1)$ in which every variable evaluates to $0$ and the target is $1$.

Apply the modification to the YES instance of Section A.1 with universe $U = \{1, \ldots, 6\}$ and the exact cover $\{S_1, S_2\}$. Each of the six original rows now appears three times, giving 18 data points, and the new point brings the total to 19. The original cover expression $f(\mathbf{x}) = x_1 + x_2$ continues to fit each duplicated row exactly. On the new point, $f(\mathbf{0}) = 0$ against target 1, contributing squared error 1. The total loss is 1.

This is optimal. With $\mathcal{P} = \{+\} \cup \{x_1, \ldots, x_4\}$, every expression is a non-negative integer combination of variables, so $f(\mathbf{0}) = 0$ for any such $f$. The new point therefore contributes at least 1 to the loss of any feasible expression, so $\mathrm{OPT}_k = 1$ on this YES instance.

For contrast, consider a NO instance: universe $U' = \{1, 2, 3\}$ with sets $S_1' = \{1, 2\}$, $S_2' = \{2, 3\}$, $S_3' = \{1, 3\}$ and $k = 2$. Every pair of these sets overlaps, so no exact cover of size two exists. The $k$-variable reduction (Theorem 2) applied to this instance yields the dataset:

| Element $e$ | $x_1$ | $x_2$ | $x_3$ | Target |
|:---:|:---:|:---:|:---:|:---:|
| 1 | 1 | 0 | 1 | 1 |
| 2 | 1 | 1 | 0 | 1 |
| 3 | 0 | 1 | 1 | 1 |

After triplicating each row and appending $(\mathbf{0}, 1)$, the dataset has 10 points. Every two-variable expression $x_i + x_j$ with $i \neq j$ miscounts exactly one element of $U'$:

| Expression | Element 1 | Element 2 | Element 3 | Original error |
|:---:|:---:|:---:|:---:|:---:|
| $x_1 + x_2$ | 1 | 2 | 1 | 1 (at element 2) |
| $x_1 + x_3$ | 2 | 1 | 1 | 1 (at element 1) |
| $x_2 + x_3$ | 1 | 1 | 2 | 1 (at element 3) |

The single miscount on the original data, once triplicated, contributes 3 to the loss, and adding the new-point error of 1 yields total loss 4. Increasing any coefficient only worsens the loss (e.g., $f = 2x_1 + x_2$ produces 2 at element 1 and 3 at element 2, costing $1 + 4 = 5$ on the original data alone). Hence $\mathrm{OPT}_k = 4 = R + 1$ on this NO instance.

The two instances now have $\mathrm{OPT}_k = 1$ and $\mathrm{OPT}_k = 4$ respectively, separated by the chosen $R = 3$. An $\alpha(k)$-FPT-approximation, given the YES instance, would output some $\hat{f}$ with $\mathcal{L}(\hat{f}, \mathcal{D}) \leq \alpha(k) \cdot \mathrm{OPT}_k = 2$ and integrality of the squared loss tightens this to $\mathcal{L}(\hat{f}, \mathcal{D}) \leq R - 1 = 2$. Given the NO instance, every feasible $\hat{f}$ has $\mathcal{L}(\hat{f}, \mathcal{D}) \geq \mathrm{OPT}_k = 4 = R + 1$. Comparing the algorithm's output loss to $R$ thus decides EXACT COVER in FPT time, contradicting W[1]-hardness.

Two features of the construction deserve note. The triplication amplifies any non-zero error on the original data by exactly the factor needed to defeat $\alpha(k)$, regardless of how fast $\alpha$ grows. The new point $(\mathbf{0}, 1)$ ensures even YES instances have $\mathrm{OPT}_k \geq 1$, ruling out a degenerate case in which a multiplicative approximation of $\mathrm{OPT}_k = 0$ is trivially zero.

### A.4   Cross-Composition from UNBOUNDED $k$-SUM

We illustrate the cross-composition from Theorem 5 with a concrete instance.

Consider three instances of UNBOUNDED $k$-SUM, each asking whether $k = 3$ elements (with repetition allowed) can be selected to sum to target $T = 10$:

$$\text{Instance 1:} \quad S_1 = \{2, 3, 5\}$$
$$\text{Instance 2:} \quad S_2 = \{1, 4, 7\}$$
$$\text{Instance 3:} \quad S_3 = \{2, 4, 6\}$$

Instance 1 is a YES instance: $2 + 3 + 5 = 10$. Instance 2 is a NO instance: no combination of three elements sums to 10 (the closest are $1 + 4 + 4 = 9$ and $1 + 4 + 7 = 12$). Instance 3 is a YES instance: $2 + 2 + 6 = 10$.

The cross-composition constructs a single SR instance that is satisfiable if and only if at least one input instance is satisfiable.

We first choose a base $B$ larger than $T + k \cdot \max_{i,j} a_{i,j} = 10 + 3 \cdot 7 = 31$. For clarity, we use $B = 100$. Each element is then tagged with a power of $B$ identifying its source instance:

| Instance | Original values $a_{i,1}$ | $a_{i,2}$ | $a_{i,3}$ | Tagged values $\tilde{a}_{i,j} = a_{i,j} + B^i$ $\tilde{a}_{i,1}$ | $\tilde{a}_{i,2}$ | $\tilde{a}_{i,3}$ |
|---|---|---|---|---|---|---|
| 1 | 2 | 3 | 5 | 102 | 103 | 105 |
| 2 | 1 | 4 | 7 | 10001 | 10004 | 10007 |
| 3 | 2 | 4 | 6 | 1000002 | 1000004 | 1000006 |

The SR instance has primitive set $\mathcal{P} = \{+\} \cup \{c_{i,j}\}$ where constant $c_{i,j}$ has value $\tilde{a}_{i,j}$, giving 10 primitives total. The dataset has three data points with targets $y_i = T + k \cdot B^i$:

| Data point | Target $y_i$ | Interpretation |
|---|---|---|
| 1 | $10 + 3 \cdot 100 = 310$ | Correct iff 3 elements from $S_1$ sum to 10 |
| 2 | $10 + 3 \cdot 10000 = 30010$ | Correct iff 3 elements from $S_2$ sum to 10 |
| 3 | $10 + 3 \cdot 1000000 = 3000010$ | Correct iff 3 elements from $S_3$ sum to 10 |

We use 0-1 loss (counting mispredictions) with threshold $\varepsilon = 2$, meaning the expression must predict at least one target correctly. The parameter is expression size $s = 2k - 1 = 5$, which corresponds to summing exactly $k = 3$ constants.

Consider the YES instance 1 with solution $2 + 3 + 5 = 10$. The corresponding expression is $f = c_{1,1} + c_{1,2} + c_{1,3}$, which evaluates to $102 + 103 + 105 = 310 = y_1$. This expression has size 5 (three constant leaves and two addition nodes), predicts $y_1$ correctly, and achieves loss 2.

Similarly, the YES instance 3 with solution $2 + 2 + 6 = 10$ corresponds to $f = c_{3,1} + c_{3,1} + c_{3,3}$ (note the repetition of $c_{3,1}$). This evaluates to $1000002 + 1000002 + 1000006 = 3000010 = y_3$.

Now consider what happens if we try to mix constants from different instances. Suppose we attempt $f = c_{1,1} + c_{1,2} + c_{2,1} = 102 + 103 + 10001 = 10206$. This equals none of the targets:

| Target | Value | Match? |
|---|---|---|
| $y_1$ | 310 | No |
| $y_2$ | 30010 | No |
| $y_3$ | 3000010 | No |

The tagging scheme prevents mixing. To see why, decompose any expression summing $k$ constants into its base-$B$ representation. If $s_i$ constants come from instance $i$, the sum is:

$$\underbrace{\sum_{\text{selected}} a_{i,j}}_{\text{untagged sum}} + \sum_{i=1}^{3} s_i \cdot B^i$$

For this to equal $y_m = T + k \cdot B^m$, the second term must equal $k \cdot B^m$. Since each $s_i \leq k < B$ and the untagged sum is less than $B$, there are no carries in base $B$. Unique representation then forces $s_m = k$ and $s_i = 0$ for $i \neq m$. All constants must come from a single instance.

The cross-composition combines $t$ instances into one SR instance with parameter $s = O(k)$. If SR admitted a polynomial kernel, we could compress arbitrarily many instances into size $\text{poly}(k)$, which would imply $\mathsf{NP} \subseteq \mathsf{coNP/poly}$. This rules out efficient preprocessing for SR.

# B   Supplementary Proofs

**Unbounded $k$-Sum.** Given a finite set $S = \{a_1, \ldots, a_n\}$ of positive integers, target $T$, and integer $k$ with $k \leq |S|$: decide if there exist $k$ elements from $S$ *with repetition allowed* that sum to $T$. Equivalently, do there exist non-negative integers $m_1, \ldots, m_n$ such that $\sum_{i=1}^{n} m_i = k$ and $\sum_{i=1}^{n} m_i a_i = T$?

**Claim 1.** UNBOUNDED $k$-SUM *remains NP-hard under the restriction $k \leq |S|$.*

*Proof.* We reduce from EXACT COVER, which is NP-complete (Karp, 1972). We may discard trivial instances with $k > m$, where $m$ is the number of sets.

**Exact Cover.** Given a universe $U = \{e_1, \ldots, e_u\}$, a set $\mathcal{S} = \{S_1, \ldots, S_m\}$ of nonempty subsets of $U$, and an integer $k$: decide whether there exist exactly $k$ sets in $\mathcal{S}$ that partition $U$ (i.e., are pairwise disjoint and cover every element).

**Construction.** Given an EXACT COVER instance $(U, \mathcal{S}, k)$, we construct an UNBOUNDED $k$-SUM instance as follows. Let $B = k + 1$.

- For each set $S_j \in \mathcal{S}$, create integer $a_j = \sum_{i:\, e_i \in S_j} B^{i-1}$, encoding membership in base $B$.
- Set target $T = \sum_{i=1}^{u} B^{i-1} = \frac{B^u - 1}{B - 1}$.
- The parameter remains $k$.

**Correctness.** We demonstrate that EXACT COVER has a solution iff the UNBOUNDED $k$-SUM instance has a solution.

($\Rightarrow$) Suppose $\{S_{j_1}, \ldots, S_{j_k}\}$ is an exact cover. Since these sets partition $U$, each element $e_i$ belongs to exactly one selected set. Therefore,

$$\sum_{\ell=1}^{k} a_{j_\ell} \;=\; \sum_{i=1}^{u} 1 \cdot B^{i-1} \;=\; T.$$

($\Leftarrow$) Suppose sets $S_{j_1}, \ldots, S_{j_k}$ (possibly with repetition) satisfy $\sum_{\ell=1}^{k} a_{j_\ell} = T$. Let $c_i$ denote the number of selected sets containing $e_i$, counting multiplicity. Then

$$\sum_{\ell=1}^{k} a_{j_\ell} \;=\; \sum_{i=1}^{u} c_i \cdot B^{i-1}.$$

Since we select exactly $k$ sets, each element can appear in at most $k$ of them, so $0 \leq c_i \leq k < B$ for all $i$. Because every digit $c_i$ is strictly less than the base $B$, no carrying occurs in the base-$B$ representation. Thus the expression $\sum_{i=1}^{u} c_i \cdot B^{i-1}$ is the unique base-$B$ expansion of the sum.

The target $T = \sum_{i=1}^{u} 1 \cdot B^{i-1}$ has all base-$B$ digits equal to 1. By uniqueness of representation, we must have $c_i = 1$ for every $i \in \{1, \ldots, u\}$. Hence each element is covered exactly once. Moreover, no set can be selected more than once: if $S_j$ were selected twice, every element in $S_j$ would satisfy $c_i \geq 2$, a contradiction. Thus the $k$ selected sets are distinct and form an exact cover.

The reduction runs in polynomial time (each $a_j$ has $\mathcal{O}(u \log B) = \mathcal{O}(u \log k)$ bits), and the constructed instance satisfies $k \leq m = |S|$, completing the proof. $\square$

