# OpenReview forum: "How Hard Is Science?"
_ICML.cc/2026/Conference — ICML 2026 regular_

### Official Review · Reviewer_Szjb · 2026-03-08

**Soundness:** 4
**Presentation:** 2
**Significance:** 3
**Originality:** 4
**Overall Recommendation:** 5
**Confidence:** 4

**Summary:**

Focusing on the discrepancy between the established theory that symbolic regression is an NP-hard problem and the numerous practical symbolic regression algorithms that can find exact formulas within a finite time, the authors propose using parametric complexity theory to analyze the complexity of symbolic regression algorithms with respect to not only the number of data sample points $n$ but also the formula size, depth, or number of variables $k$. The authors prove that, when the depth or size of the formula trees is restricted, the size of the search space is independent of the number of samples $n$, such that the operation of enumerating all formulas in the search space and evaluating them individually has polynomial complexity with respect to $n$. However, when the number of variables or primitives is limited to $k$, the complexity of evaluating each formula changes from a polynomial of $n$ to $n^{O(k)}$, which makes the problem an intractable W[1]-Hardness problem. The authors further find lower bounds under the exponential time hypothesis, prove approximation hardness, and rule out polynomial kernels.

**Compliance With Llm Reviewing Policy:**

Affirmed.

**Final Justification:**

I appreciate the authors' efforts in the rebuttal, which addressed my concerns about the manuscript's significance and convinced me that parametric complexity theory can not only give some obvious conclusions presented in the current manuscript but also provide constructive guidance for more specific problems. We hope the authors can provide more toy examples in the revised manuscript (perhaps in the Appendix) illustrating how to theoretically evaluate the complexity of different types of symbolic regression algorithms (e.g., tree-based or DAG-based) through mathematical derivation, thus providing valuable theoretical guidance for researchers in the field of symbolic regression to design new algorithms.

**Key Questions For Authors:**

1. What is the specific definition of Tractable? Why is $n^{O(1)}$ in FPT Tractable, while $n^{O(k)}$ in $W[1]$-hardness is intractable?

2. Ref [1] proposes a symbolic regression algorithm for discovering network dynamics formulas, which allows the discovery of network dynamics formulas describing the interaction and evolution of variables on hundreds of nodes by introducing network dynamics operators. Does this contradict the conclusion in this paper that Variable Number leads to intractable $W[1]$-hardness?

3. Can this set of parameterized complexity theories be used to analyze expected complexity, i.e., the average computational cost of a specific heuristic search algorithm finding the optimal formula with probability $P$ on a loss landscape generated by a specific data distribution?

[1] Discovering network dynamics with neural symbolic regression (Nature Computational Science, 2025)

**Limitations:**

In the Limitation Section, the authors acknowledge the manuscript's limitations regarding data noise, constant optimization, and Worst-Case Complexity. However, as discussed in W2, the limitations of Worst-Case Complexity warrant further discussion, such as whether classical parametric complexity theory is sufficient to analyze the expected complexity of an inherently stochastic algorithm like heuristic symbolic regression.

**Strengths And Weaknesses:**

S1. (Originality & Significance) The article focuses on the discrepancy between the theoretical hardness of symbolic regression and its practical feasibility, which would be a topic of interest to many researchers in symbolic regression.

S2. (Soundness) Using parametric complexity theory and rigorous derivation, the authors explain how restricting the size or depth of the formula in practice can transform symbolic regression from NP-hard to tractable.

W1. (Presentation) This article is not easy to read for those unfamiliar with parametric complexity theory. First, terms such as FPT that appear in the Introduction Section are not fully explained until Section 3. Furthermore, the key term "tractable" is not given a specific definition (why FPT means tractable, but W[1]-Hardness does not?). In addition, a reader unfamiliar with the complexity theory of symbolic regression may consider the input size $n$ as the size constraint of the formula rather than the number of data points, since intuitively the former is the core factor affecting the size of the search space and determining the tractability of the symbolic regression algorithm.

W2. (Significance) The authors focus on worst-case complexity and, based on this, define the complexity of symbolic regression as the computational cost required to traverse and evaluate every formula in the search space. However, this is not how symbolic regression algorithms actually work; they typically perform a heuristic search on the loss landscape generated by the data samples. Therefore, rather than the computational cost of traversing the entire space and evaluating formula losses, researchers in symbolic regression might be more concerned with the computational cost required to discover the target formula with probability $P$ with respect to the specific heuristic search algorithm on the specific landscape provided by the data distribution.

---

> ### Author Rebuttal · Authors · 2026-03-30
>
> We thank the reviewer for the careful reading and positive assessment of the paper's originality and soundness.
>
> > [W1] - (Presentation) This article is not easy to read [...]
>
> This feedback is very helpful. We will add an intuitive preview of definitions (FPT, W[1]-hard, tractable) to the introduction so readers can follow the high-level claims before the formal treatment in Section 3. We can also add wording to make more clear what is meant by the terms used in the introduction. This would include clarifying the definition of tractability and what is being taken as the input size, as mentioned in your comment.
>
> > [W2] - However, this is not how symbolic regression algorithms actually work; they typically perform a heuristic search on the loss landscape generated by the data samples.
>
> This is a well-taken point and reflects a genuine limitation of the worst-case framework. It is worth noting that some work [1] has explored exhaustive search in the tractable regime, so our results do have some direct practical applicability there but the broader point about heuristic complexity stands, and we address it further below.
>
> [1] https://doi.org/10.1109/TEVC.2023.3280250
>
> > [W2] - Therefore, rather than [...] by the data distribution.
>
> This is an interesting point. We mention that worst-case analysis is a limitation in Section 8. Overall, it is still meaningful since, for the unrestricted worst-case problem, efficient algorithms with guarantees better than what we show cannot be obtained under the standard assumptions. But we agree that for applications we would like to include some heuristic distribution and make some assumptions about that.
>
> Analyzing the expected complexity of a specific heuristic (e.g., the cost for a GP to find the target formula with probability $P$ on a given data distribution) is interesting and distinct from parameterized worst-case analysis. We can add this to §9 (open questions).
>
> > [Q1] - What is the specific definition of Tractable? Why is f(k) * n^O(1) in FPT Tractable, while n^O(k) in W[1]-hardness is intractable?
>
> In the FPT regime, runtime is $f(k) \cdot n^{O(1)}$ so once $k$ is fixed, the problem is polynomial in $n$ with a degree that does not depend on $k$. In the W[1]-hard regime, what is ruled out (under standard assumptions) is an algorithm of that form $f(k) \cdot n^{O(1)}$. By contrast, runtimes like $n^{O(k)}$ are XP. This means that they are polynomial for each fixed $k$, but with a degree that grows with $k$, so even modest values of $k$ can make them infeasible for large $n$ (e.g., $k = 30$ gives $n^{30}$). The distinction is whether the parameter can be factored out of the polynomial dependence on data size. We can add some additional text to the introduction to make this more clear. Thanks for pointing this out.
>
> > [Q2] - Does this contradict the conclusion in this paper that Variable Number leads to intractable hardness?
>
> This was an interesting reference; thanks for sharing it. The "hundreds of nodes" in ND² is the network size, not the number of freely selectable variables in a single unrestricted SR instance.
>
> ND² searches for shared network-dynamics laws using strong domain structure. In particular, it reduces searches on high-dimensional networks to equivalent one-dimensional systems and works with a specialized node/edge dynamics formulation rather than unrestricted variable selection.
>
> So we do not view this as contradicting our result. Rather, it illustrates that domain knowledge and a highly structured search space can avoid the unrestricted variable-selection bottleneck identified by our W[1]-hardness result.
>
> > [Q3] - Can this set of parameterized complexity theories [...] specific data distribution?
>
> Yes, in principle. Parameterized complexity theory is essentially just complexity theory with an additional parameter $k$ derived from the problem so there’s nothing that restricts it to worst-case analysis. One could define average-case or distributional variants, analyzing the expected cost of a specific heuristic on a specific loss landscape, parameterized by expression depth, variable count, etc.
>
> To expand on the latter question, if we knew the exact loss landscape and heuristic, or at least some broad properties of them, yes but that would limit the results to those particular assumptions. We didn't do this because that would limit the generality and picking the right level of abstraction to answer these sorts of questions is difficult. We agree that it’s important to move towards this kind of analysis and list it in the future work section for this reason.
>
> > However, as discussed in W2, the limitations of Worst-Case Complexity warrant further discussion [...] like heuristic symbolic regression.
>
> Hopefully this is addressed by our reply above. We can add additional language to Section 8 to make this more clear. Thanks for pointing this out.

---

> > ### Author Rebuttal · Reviewer_Szjb · 2026-04-01
> >
> > My questions have been largely answered, and I am pleased to see that the author is willing to provide further discussion of the proposed issues in their revised version. However, my particular concern (W2) still did not receive a very convincing response. Although the authors used the work "Exhaustive Symbolic Regression" to illustrate the practical value of their analysis, these "worst-case" works are usually limited to a single / few variable (Exhaustive Symbolic Regression) or short formulas with length lower than 8 (POVE: A Preoptimized Vault of Expressions for Symbolic Regression Research and Benchmarking). Although the authors stated in their reply to Q3 that analysis is possible in principle, they did not offer any intuition or insight into how to conduct the analysis (possibly due to 5000 character limits). A simple example that can systematically analyze the performance difference between two different kind of symbolic regression algortihms (e.g., symbolic tree-based vs. DAG-based, GP-based vs. MCTS-based, DL-based vs. RL-based, etc) would be extremely convincing.

---

> > > ### Author Response · Authors · 2026-04-03
> > >
> > > Thank you for clarifying the sort of example you had in mind. Below, we sketch a toy comparison, along the lines you suggested, of tree-based versus DAG-based symbolic regression under the same size-bounded best-improvement search rule. For brevity, we make some simplifying assumptions.
> > >
> > > Fix the primitive set $\mathcal P$ = {$\times,x,1$}. For each $h\ge 0$, let the target formula be
> > > $$
> > > f_h(x)=x^{2^h}.
> > > $$
> > >
> > > Here $h$ is not the algorithmic parameter; it simply indexes a family of targets of increasing structural complexity. The parameterized quantity is the structural budget $k$: for tree-based SR, $k$ is the allowed tree size, while for DAG-based SR, $k$ is the allowed DAG size.
> > >
> > > Use the two-point dataset
> > > $$
> > > \mathcal D_h=\{(2,2^{2^h})\,(3,3^{2^h})\}
> > > $$
> > >
> > > with squared loss. We fix the structural budget at $k=h+1$ and ask whether each representation can recover $f_h$ within this budget. Consider two algorithms sharing the same primitive set, the same dataset, and the same greedy objective: start from $x$, and at each step apply the loss-minimizing single-gate expansion permitted by the chosen representation, subject to the size budget $k$. They differ only in representation: one searches formula trees, while the other searches expression DAGs, so repeated subexpressions may be shared only in the latter.
> > >
> > > This can be analyzed exactly. In the DAG representation, define
> > > $$
> > > g_0=x,\qquad g_{i+1}=g_i\cdot g_i.
> > > $$
> > >
> > > Then $g_h=f_h$, using one leaf plus $h$ multiplication nodes, so the minimum exact DAG size is
> > > $$
> > > k_{\mathrm{DAG}}(h)=h+1.
> > > $$
> > >
> > > Moreover, since $\times$ is the only operator, every candidate is a monomial $x^m$. If the largest exponent currently present is $2^i$, then every one-step candidate satisfies $m\le 2^{i+1}$. On $\mathcal D_h$, the squared loss is strictly decreasing in $m$ for $m\le 2^h$, so the best move is always to square the current largest node. The DAG algorithm therefore follows
> > > $$
> > > x,\ x^2,\ x^4,\ \dots,\ x^{2^h},
> > > $$
> > >
> > > so its hitting time is
> > > $$
> > > T_{\mathrm{DAG}}=h,
> > > $$
> > >
> > > and $\Pr[T_{\mathrm{DAG}}\le t]=1$ for all $t\ge h$.
> > >
> > > For trees, repeated subexpressions cannot be shared. If $s(h)$ denotes the minimum exact tree size of $x^{2^h}$, then
> > > $$
> > > s(0)=1,\qquad s(h+1)=2s(h)+1,
> > > $$
> > >
> > > so
> > > $$
> > > k_{\mathrm{tree}}(h)=2^{h+1}-1.
> > > $$
> > >
> > > Therefore, under the same budget $k=h+1$, the tree-based algorithm cannot represent the target at all, so $\Pr[T_{\mathrm{tree}}\le t]=0$ for all $t$.
> > >
> > > Thus the family $\{f_h\}$ yields a concrete parameterized comparison between two SR algorithm families: with the same primitive set, the same dataset, and the same greedy objective, DAG-based search reaches the target with linear budget and linear hitting time, whereas tree-based search requires exponentially larger structural budget before exact recovery is possible. This illustrates the same structural-parameter viewpoint as our main results: the effective size parameter depends strongly on the chosen representation. Although this toy analysis uses a deterministic greedy rule for clarity, the hitting-time guarantee it yields aligns with the kind of per-instance cost analysis described in W2.
> > >
> > > We hope this gives a sense of how such an analysis could be extended from our results.

---

### Official Review · Reviewer_MTok · 2026-03-11

**Soundness:** 3
**Presentation:** 4
**Significance:** 3
**Originality:** 4
**Overall Recommendation:** 4
**Confidence:** 3

**Summary:**

This paper examines the computational hardness of Symbolic Regression (SR) through the lens of parameterized complexity theory. While SR is known to be NP-hard, practical SR algorithms discover accurate and interpretable models in reasonable time. To explain this gap between worst-case theory and practical success, the authors systematically investigate the complexity of SR under various parameterizations. The main results include:
1. SR is fixed-parameter tractable (FPT) when parameterized by expression depth or tree size (Theorem 1).
2. SR becomes W[1]-hard when parameterized by the number of variables or primitives used (Theorem 2).
3. Lower bounds under the Exponential Time Hypothesis for selection-type parameterizations (Theorem 3).
4. Inapproximability of loss minimization (Theorem 4).
5. The non-existence of polynomial kernels (Theorem 6).

**Compliance With Llm Reviewing Policy:**

Affirmed.

**Final Justification:**

This paper provides a systematic parameterized complexity analysis of Symbolic Regression. The paper is well-organized and clearly written.

A limitation of my review is that I am not an expert in parameterized complexity theory and was unable to verify the proofs in detail. I maintain my recommendation of **weak accept (4)**. The paper makes a theoretical contribution to understanding SR complexity with practical relevance. Its impact may be somewhat specialized to the SR theory communities.

**Key Questions For Authors:**

1. Theorem 2 uses only addition and variables as the primitive set, so the resulting expressions are just linear sums. Is this W[1]-hardness result merely reflecting the known difficulty of linear regression, rather than a combinatorial difficulty for SR?

**Limitations:**

yes

**Strengths And Weaknesses:**

**Strengths**:

1. **Theoretical contributions.**
	- The FPT results explain why practical algorithms such as PySR and Operon succeed by bounding expression depth or node count.
	- The W[1]-hardness results pinpoint feature selection as the computational bottleneck in SR, which is consistent with empirical observations. PySR's documentation notes that symbolic regression works well on low-dimensional datasets, and this paper provides complexity-theoretic justification for that empirical finding.
2. **Practical implications.** The discussion in Section 8 effectively connects theoretical results to the design choices of real SR algorithms. The FPT results correspond to the common practice of bounding node count (typically $\leq 30$) or depth ($\leq 6$).

**I am not an expert in the theoretical analysis of symbolic regression, and I am not familiar with the parameterized complexity framework used in this paper.** Overall, the paper is well-organized and Table 1 provides a clear at-a-glance overview of results.

---

> ### Author Rebuttal · Authors · 2026-03-30
>
> We thank the reviewer for their positive assessment and the focused technical question.
>
> > [Q1] - Theorem 2 uses only addition and variables as the primitive set, so the resulting expressions are just linear sums. Is this W[1]-hardness result merely reflecting the known difficulty of linear regression, rather than a combinatorial difficulty for SR?
>
> The reviewer raises an important distinction. Best Subset Selection (i.e., finding $k$ variables from $m$ candidates whose linear combination fits the data) is known to be NP-hard (Natarajan, 1995). However, NP-hardness alone does not settle the parameterized picture since NP-hard problems (e.g., Vertex Cover) can still be FPT, meaning they admit algorithms with runtime $f(k) \cdot \text{poly}(n)$ where $k$ is isolated from the exponent of $n$. An NP-hard problem could still have an $n^{O(k)}$ algorithm that is polynomial for each fixed $k$. Theorem 2 establishes that $k$-Variable SR is W[1]-hard, which is stronger from a parameterized point of view. Under standard assumptions, no algorithm can achieve runtime $f(k) \cdot \text{poly}(n)$.
>
> The hardness in Theorem 2 comes from variable selection, which is the same combinatorial problem underlying Best Subset Selection in linear regression, as the reviewer suggests. This shows that even when the expression language is simple (just addition), the variable selection bottleneck already makes SR W[1]-hard. The theorem shows this hardness is intrinsic to the selection step, not an artifact of expressive primitives.
>
> We're glad this came up, it's an important point that we'll add to the paper.
>
> ---
>
> Hopefully this clarifies the result. We’re happy to answer any further questions in the discussion period.

---

> > ### Author Rebuttal · Reviewer_MTok · 2026-04-01
> >
> > I thank the authors for the informative response to my question. Overall, the rebuttal has addressed my question (Q1). I maintain my recommendation of **weak accept (4)** and raise my confidence from **2** to **3**.

---

### Official Review · Reviewer_iyAZ · 2026-03-11

**Soundness:** 3
**Presentation:** 3
**Significance:** 2
**Originality:** 2
**Overall Recommendation:** 4
**Confidence:** 4

**Summary:**

Parameterized complexity analysis of symbolic regression (SR). SR is FPT when parameterized by expression depth or tree size (Theorem 1, Corollary 1), W[1]-hard when parameterized by number of variables or primitives used (Theorem 2, Corollary 2). Additional ETH lower bounds, inapproximability, and kernelization lower bounds.

**Compliance With Llm Reviewing Policy:**

Affirmed.

**Key Questions For Authors:**

See above.

**Limitations:**

Section 8.3 discusses noise, constants, and worst-case nature. Adequate but brief. The scope mismatch between the kernelization result (growing P) and the FPT results (fixed P) is not acknowledged. The gap between exhaustive enumeration and practical GP-based search could be explored more.

**Strengths And Weaknesses:**

Clean dichotomy between structural parameters (depth/size, FPT) and selection parameters (variables/primitives, W[1]-hard). Identifying feature selection as the hardness source (Remark 2) is the most useful insight. Reduction from Exact Cover (Theorem 2) is well-constructed. ETH lower bounds (Theorem 3) nearly match brute-force. Paper well-organized, Table 1 helpful, worked examples in Appendix A help verification.

The FPT algorithm (Theorem 1) is exhaustive enumeration over all bounded-depth expressions. Technically correct but not algorithmically interesting. Practical SR systems use GP-based heuristics taht do not enumerate, so the claimed "explanation" of practical success is limited to showing the problem class is tractable, not that practical algorithms are justified. Authors should clarify this distinction.

Definition 1 includes variables in the primitive set P. When Theorem 1 says "fix P," this fixes the variable count d. If d grows with input (the typical SR setting), the result is XP, not FPT in depth alone. This is central to the paper's message and needs explicit clarification.

Theorem 6 (no polynomial kernel) has a scope mismatch with the FPT results. The cross-composition constructs a primitive set that grows with the number of instances, while the FPT results assume fixed P. The kernel lower bound thus applies to a different problem formulation than the one shown to be FPT. Authors should clarify whether a kernel lower bound can be established for fixed P.

Treatment of constants is thin. Practical SR optimizes continuous constants; the paper's results apply to a discrete version. It is unclear whether the FPT results survive if constants are treated as continuous parameters optimized per candidate structure.

Inapproximability Theorem 4 relies on multiplicative approximation with OPT possibly 0. Any finite ratio trivially fails when OPT=0. Technically correct but not very informative. Additive or promise-gap versions would be more meaningful.

The arithmetic cost model in Theorem 1 deserves attention. Remark 1 assumes constant-time primitive evaluation, but intermediate values can grow in bit-length with depth (e.g., repeated multiplication). This can be repaired by adopting a real-RAM model or bounding numeric growth, but should be stated explicitly.

---

> ### Author Rebuttal · Authors · 2026-03-31
>
> We thank the reviewer for their careful and technically detailed reading. These comments were helpful and have improved the paper.
>
> We will clarify throughout that Theorem 1 only shows tractability of the bounded-complexity problem class and does not by itself justify GP-style heuristics. There is some related exhaustive-search work in the small-parameter regime [1], so our results do connect directly to practice there, but the gap to heuristic search remains important and we will discuss it more explicitly.
>
> [1] https://doi.org/10.1109/TEVC.2023.3280250
>
> On fixed $\mathcal{P}$ vs. growing $d$: the reviewer is correct that because variables are elements of $\mathcal{P}$, Theorem 1 fixes $d$ as well. If $d$ grows with the input, the runtime becomes $d^{O(a^k)} \cdot \mathrm{poly}(n)$, which is XP rather than FPT in depth alone. We will make this explicit after Theorem 1 and throughout the paper. The intended FPT regime is the one where the primitive library (including variables) is chosen ahead of time and only the dataset size $n$ grows.
>
> On kernelization: thanks for pointing this scope mismatch out. The reviewer is right that Theorem 6 applies to a different formulation. A uniform kernel lower bound for all fixed $\mathcal{P}$ is false: for $\mathcal{P}=\{+\}\cup X\cup C$ with $X,C$ fixed finite leaf sets, every size-$s$ expression is just a multiset sum over a constant-size leaf set, so there are only $s^{O(1)}$ candidates. For polynomial-time-computable losses, SR-DEC is then in $\mathsf{P}$ and has a polynomial kernel. Thus Theorem 6 is a kernel lower bound for the $\mathcal{P}$-as-input formulation, not for the fixed-$\mathcal{P}$ regime of Theorem 1. For richer fixed bases (e.g., $\{+,\times,0,1,x_1,\ldots,x_d\}$ with $d$ fixed), the kernelization complexity remains open, and we will say so explicitly. The wording will be updated throughout to make these distinctions more precise.
>
> On constants: the hardness and inapproximability results are unchanged because the reductions use no constants. For the FPT results, if fitting continuous constants for a candidate structure of parameter at most $k$ can be done in $g(k)\cdot\mathrm{poly}(n)$ time, then the enumeration-based FPT result survives with that optimizer in the inner loop; if not, this falls outside the FPT regime. We will add a forward reference from Remark 1 to §6.2.
>
> On Theorem 4: this was a helpful suggestion that led us to a stronger result. For the exact-$k$ variable version, we can add one unfittable point $(\mathbf{0},1)$. Then YES instances have $\mathrm{OPT}_k=1$, while NO instances have $\mathrm{OPT}_k\ge 2$, since loss 1 would force zero loss on all original points, i.e. an exact cover of size exactly $k$. Hence distinguishing $\mathrm{OPT}_k\le 1$ from $\mathrm{OPT}_k\ge 2$ is W[1]-hard, and there is no $(2-\varepsilon)$-FPT-approximation unless FPT = W[1]. The same conclusion transfers to the primitive-count formulation via the same one-step reparameterization as in Corollary 2. Thanks for this idea; we think it improves the paper. Hopefully it also addresses your concern.
>
> On arithmetic cost: we agree this should be stated explicitly. We will revise Remark 1 to adopt a real-RAM / unit-cost arithmetic model (or equivalently state bounded numeric-growth assumptions) rather than leaving this implicit. We will also expand §8.3 slightly in the final version. Thanks for catching this.

---

> > ### Author Rebuttal · Reviewer_iyAZ · 2026-04-01
> >
> > The authors addressed all concerns raised in my review. The clarification on the FPT/XP distinction when d grows with input, the kernelization scope mismatch, and the improved inapproximability result (promise-gap version) are all satisfactory. I maintain my score.

---

### Official Review · Reviewer_Rg2Z · 2026-03-13

**Soundness:** 3
**Presentation:** 4
**Significance:** 3
**Originality:** 3
**Overall Recommendation:** 5
**Confidence:** 3

**Summary:**

This work studies the computational complexity of symbolic regression from the perspective of parameterized complexity theory. This work aims to explain why practical systems often succeed despite the problem being NP-hard.

They show that SR becomes FPT when parameterized by expression depth or tree size, meaning that bounded-complexity expressions can be searched efficiently. The problem becomes W[1]-hard when parameterized by the number of variables or primitives used, identifying feature/primitive selection as a key source of computational difficulty.

The paper also establishes ETH-based lower bounds and shows that SR admits no polynomial kernel under common parameterizations. These results theoretically explain why SR methods work well when expression complexity is constrained but struggle with high-dimensional feature selection.

**Compliance With Llm Reviewing Policy:**

Affirmed.

**Key Questions For Authors:**

1. I did a rough reading of the proofs. The theoretical formulation assumes constants come from a finite set, while practical symbolic regression systems typically treat constants as continuous parameters optimized after the expression structure is chosen.
Will the complexity results change under this more realistic model with continuous constant optimization?

2. Does this result suggest that separating feature selection from expression search could significantly improve symbolic regression pipelines?

**Strengths And Weaknesses:**

### Strengths

1. The complexity analysis looks solid. It basically reached a conclusion: SR is fixed-parameter tractable when parameterized by expression depth or tree size, and it becomes W[1]-hard when parameterized by number of variables or number of primitives. Basically, the main source of hardness is primitive selection.
2. The results align well with empirical practices in modern symbolic regression systems.
3. Paper presentation is clear.

### Weaknesses

1. The FPT algorithms for bounded depth or size essentially rely on enumerating all candidate expression trees and evaluating them on the dataset. This mainly formalizes what is already implicitly assumed in bounded search, so I think it is somewhat trivial as one of the contributions.
2. There is still a common issue in SR complexity analysis. The hardness results rely on reductions such as EXACT COVER. While this is standard in complexity theory, such constructions may not reflect the structure of real-world datasets used in symbolic regression, such as scientific discovery problems. I saw a similar open problem discussed in the conclusion.
3. Although not critical for this paper, it would be great to see experiments evaluating whether the theoretically tractable regimes correspond to practical runtime behavior in existing symbolic regression systems.

---

> ### Author Rebuttal · Authors · 2026-03-30
>
> We thank the reviewer for their thorough reading and positive recommendation.
>
> > [W1] - The FPT algorithms for bounded depth or size essentially rely on enumerating all candidate expression trees and evaluating them on the dataset. This mainly formalizes what is already implicitly assumed in bounded search, so I think it is somewhat trivial as one of the contributions.
>
> We agree the enumeration algorithm is not particularly exciting. The contribution is the proof that, when the primitive set is fixed, runtime is $f(k) \cdot \mathrm{poly}(n)$ rather than having the parameter appear in the exponent of $n$. This is meaningful even for a problem already solved heuristically. It’s analogous to proving P-membership for a problem whose polynomial-time algorithm was already informally known. We will add some language clarifying this.
>
> The result is included to contrast with the later pieces (such as the W[1] parameterization), and because that contrast is one of the main conceptual points of the paper.
>
> > [W2] - The hardness results rely on reductions such as EXACT COVER. While this is standard in complexity theory, such constructions may not reflect the structure of real-world datasets used in symbolic regression, such as scientific discovery problems. I saw a similar open problem discussed in the conclusion.
>
> We agree this is important. It requires more framework development to approach but, as you mention, we list it as an open problem (albeit briefly due to space) in section 9. Much of the time we probably want some sort of “smoothness” constraint.
>
> > [W3] - Although not critical for this paper, it would be great to see experiments evaluating whether the theoretically tractable regimes correspond to practical runtime behavior in existing symbolic regression systems.
>
> There is related work [1] on exhaustive symbolic regression in this sort of bounded-complexity regime. We will include some discussion of this work and its relation to our results in the final version.
>
> [1] https://doi.org/10.1109/TEVC.2023.3280250
>
> > [Q1] - Will the complexity results change under this more realistic model with continuous constant optimization?
>
> We discuss this in section 6.2. Essentially, some results hold regardless of the constant model, while the applicability of other results will depend on the optimization method chosen when switching to a continuous constant model.
>
> > [Q2] - Does this result suggest that separating feature selection from expression search could significantly improve symbolic regression pipelines?
>
> We think so. This is mentioned in the discussion section under "The Feature Selection Bottleneck."

---

> > ### Author Rebuttal · Reviewer_Rg2Z · 2026-04-02
> >
> > The authors addressed all of my concerns. I will maintain my score.

---

### Decision · Program_Chairs · 2026-04-30

**Decision:**

Accept (regular)

**Comment:**

This paper on the parameterized complexity of symbolic regression has received four reviews, with overall recommendations varying from "weak accept" to "accept." All reviewers acknowledge the theoretical significance of the results, as well as their technical quality and practical implications. The rebuttal phase was extensive, giving the authors the opportunity to clarify several technical issues. For these reasons, I recommend acceptance of the paper.